# D²-GCN: DATA-DEPENDENT GCNs FOR BOOSTING BOTH EFFICIENCY AND SCALABILITY

## ABSTRACT

Graph Convolutional Networks (GCNs) have gained an increasing attention thanks to their state-of-the-art (SOTA) performance in graph-based learning tasks. However, their sheer number of node features and large adjacency matrix limit their deployment into real-world applications, as they impose the following challenges: (1) prohibitive inference cost, especially for resource-constrained applications and (2) low trainability of deep GCNs. To this end, we aim to develop low-cost GCNs with improved trainability, as inspired by recent findings in deep neural network optimization which show that not all data/(model components) are equally important. Specifically, we propose a Data-Dependent GCN framework dubbed D²-GCN which integrates data-dependent dynamic skipping at multiple granularities: (1) node-wise skipping to bypass aggregating features of unimportant neighbor nodes and their corresponding combinations; (2) edge-wise skipping to prune the unimportant edge connections of each node; and (3) bit-wise skipping to dynamically adapt the bit-precision of both the node features and weights. Our D²-GCN is achieved by identifying the importance of node features via a low-cost indicator, and thus is simple and generally applicable to various graph-based learning tasks. Extensive experiments and ablation studies on 6 GCN model and dataset pairs consistently validate that the proposed D²-GCN can (1) largely squeeze out unnecessary costs from both the aggregation and combination phases (e.g., reduce the inference FLOPs by $\downarrow 1.1\times \sim \downarrow 37.0\times$ and shrink the energy cost of GCN inference by $\downarrow 1.6\times \sim \downarrow 8.4\times$), while offering a comparable or an even better accuracy (e.g., $\downarrow 0.5\% \sim \uparrow 5.6\%$); and (2) help GCNs to go deeper by boosting their trainability (e.g., providing a $\uparrow 0.8\% \sim \uparrow 5.1\%$ higher accuracy when increasing the model depth from 4 layers to 64 layers) and thus achieving a comparable or even better accuracy of GCNs with more layers over SOTA techniques (e.g., a $\downarrow 0.4\% \sim \uparrow 38.6\%$ higher accuracy for models with 64 layers). All the codes and pretrained models will be released upon acceptance.

## 1 INTRODUCTION

Graph Convolutional Networks (GCNs) have drawn increasing attention thanks to their performance breakthroughs in graph-based learning tasks. In particular, the success of GCNs are attributed to their excellent capability to learn from non-Euclidean graph structures with irregular graph neighborhood connections via two execution phases: (1) *aggregation*, during which the features from the neighbor nodes are aggregated, and (2) *combination*, in which further updates of the features of each node are made via feed-forward layers to extract more useful features.

Despite their promising performance, the unique structure of GCNs imposes prohibitive challenges for applying them to more extensive real-world applications especially those with large-scale graphs. First, GCNs' prohibitive inference cost limits their deployment into resource-constrained devices. For example, a 2-layer GCN model requires 19 Giga (G) Floating Point Operations (FLOPs) to process the Reddit graph (Tailor et al., 2021) and a latency of $2.94\times10^5$ milliseconds, when being executed on an Intel Xeon E5-2680 CPU platform (Geng et al., 2020), which is $2\times$ and $5000\times$ over that of a 50-layer powerful Convolutional Neural Network (CNN) (Awan et al., 2017), ResNet-50, respectively; Second, while CNNs with more layers are known to consistently favor a higher accuracy (Belkin et al., 2019), deeper GCNs suffer from accuracy drops compared with shallower ones (Kipf & Welling, 2016), making it difficult to unleash their full potential.

Aiming at tackling both of the aforementioned challenges, we draw inspirations from recent works. First, previous CNN works (Katharopoulos & Fleuret, 2018; Johnson & Guestrin, 2018; Coleman et al., 2018) show that not all samples are equally important during training and training on more informative samples can improve the model accuracy, motivating us to consider allocating GCN computational budgets adapting to the sample complexity. In addition, (Zhang et al., 2019) finds that not all the CNN layers equally contribute to the final model accuracy and (Wang et al., 2018) demonstrates that skipping some of the layers even helps boost the accuracy while reducing the inference cost of CNNs. Meanwhile, recent GCN works show that not all the nodes contribute equally to the feature extraction (Veličković et al., 2018) and some neighbor nodes can be randomly abandoned without affecting the task performance (Hamilton et al., 2017).

The above prior arts motivate us to consider data-dependent dynamic GCNs for (1) pushing forward their achievable accuracy-efficiency frontier and (2) improving the trainability of deeper GCNs. To this end, we adopt a new perspective as compared to existing GCN compression works and explore data-dependent dynamic GCNs on top of SOTA GCNs. Specifically, we identify the potential data-dependent patterns that are unique to GCNs at different granularitis, and then leverage them to largely squeeze out unnecessary costs within GCNs to boost their inference efficiency and trainability. Specifically, we make the following contributions:

- We propose a Data-Dependent GCN framework dubbed $D^2$-GCN, the first dynamic inference framework dedicated to GCNs. $D^2$-GCN integrates data-dependent dynamic skipping at multiple granularities: *node-wise*, *edge-wise*, and *bit-wise*, via a low-cost indicator to notably reduce the GCN inference cost, while offering a comparable or even better accuracy.

- $D^2$-GCN is found to naturally alleviate the over-smoothing issue in GCNs and thus improves the trainability of deeper GCNs, which we conjecture is because $D^2$-GCN introduces more flexibility into the models. Hence, $D^2$-GCN opens up a new knob to not only boost GCNs' inference efficiency but also provide a promising perspective towards deeper and more powerful GCNs.

- Extensive experiments and ablation studies on top of various SOTA GCNs and datasets consistently validate the effectiveness and advantages of the proposed $D^2$-GCN. In particular, $D^2$-GCN can achieve $\downarrow 1.1 \times \sim \downarrow 37.0 \times$ inference FLOPs reduction and $\downarrow 1.6 \times \sim \downarrow 8.4 \times$ lower energy cost, while leading to a comparable or even better accuracy ($\downarrow 0.5\% \sim \uparrow 5.6\%$).

## 2 RELATED WORKS

**Graph Convolutional Networks.** GCNs are one of the most widely adopted algorithms for non-Euclidean and irregular graph structures (Wu et al., 2020) to categorize the nodes in the same graph (Sen et al., 2008) or predict the class of graphs (Hu et al., 2020). They can mainly be divided into two categories: spectral-based (Kipf & Welling, 2016) and spatial-based (Gao et al., 2019). For the spectral-based GCNs, graph convolution which is based on the spectral graph theory (Chung & Graham, 1997) is firstly proposed by (Bruna et al., 2013) and improved in (Kipf & Welling, 2016; Defferrard et al., 2016; Li et al., 2018b) for wider applications and better accuracy. Meanwhile, the spatial-based GCNs (Hamilton et al., 2017) directly perform the convolution in the graph domain by aggregating the neighbor nodes' features and recent works further improve their accuracy via clustering-based graph sampling (Chiang et al., 2019), more expressive aggregation scheme (Zeng et al., 2019), and attention mechanism (Veličković et al., 2018). Orthogonal to those prior works, we explore and develop data-dependent dynamic GCNs on top of SOTA spatial-based GCNs for improving their efficiency and scalability.

**Efficient GCNs.** Motivated by the fact that the prohibitive computational cost and memory usage of GCNs, which will even expeditiously increase with the graph size, limit the development of more powerful GCNs and their deployment into real-world applications, various compression techniques has been developed, which mainly fall into three categories: pruning graphs (i.e., simpler graphs), pruning weights (i.e., sparser weights), and quantization (i.e., lower bit-precision for hidden features and weights). For pruning graphs, (Li et al., 2020b) introduces a sparse regularizer for pruning the graph connections (i.e., the graph adjacency matrix) and leverages an alternating direction method of multipliers (ADMM) training method to make the regularization differentiable; for pruning weights, (Chen et al., 2021) prunes the graph adjacency matrix and the model weights simultaneously, generalizing the lottery ticket hypothesis (Frankle & Carbin, 2018) to GCNs; for

quantization, (Tailor et al., 2021) for the first time trains GCNs with 8-bits integer arithmetic forwarding without sacrificing the classification accuracy. Our proposed D$^2$-GCN considers an unexplored and orthogonal perspective, and explores data-dependent dynamic GCNs on multiple granularity levels for achieving better accuracy-efficiency trade-offs and improving GCNs' scalability.

**Deeper GCNs.** One challenge in designing GCNs is their scalablity of deeper GCNs for (1) handling real-world large graphs and (2) unleashing the potential of more sophisticated GCN architectures, motivating various techniques along this direction. A pioneering work (Kipf & Welling, 2016) attempts to build deeper GCNs through a residual mechanism, and finds that such a design is only effective for GCNs with no more than 2 layers. Then, (Li et al., 2018a) argues that the over-smoothing issue (i.e., connected nodes in deeper layers have more similar hidden features) prevent GCN architectures from going deeper, and thus, several following works strive to design deeper GCNs by alleviating this issue. For example, (Rong et al., 2020) proposes to tackle the over-smoothing issue by randomly removing a certain number of edges from the input graph at each training epoch; (Li et al., 2019b;a) further explore a generalized aggregation function and normalization layers to boost the performance of GCNs on large-scale graph learning tasks. Our proposed D$^2$-GCN distinguishes itself as the first to explore deeper GCNs from the data-dependent aspect by incorporating automated data-dependent gating functions to alleviate the over-smoothing issue and facilitate deeper GCNs.

**Dynamic inference.** Dynamic inference methods have been developed in the context of CNNs for adapting model complexity to input data for reducing overall average inference costs. Early works (Teerapittayanon et al., 2016; Huang et al., 2017) equip DNNs with extra branch classifiers to enforce a portion of inputs to exit at earlier branches. Later works incorporate a finer-grained layer-wise skipping policy via selectively executing a subset of layers conditioned on each input data. In particular, SkipNet (Wang et al., 2018) adopts reinforcement learning to learn the layer-wise skipping policy and BlockDrop (Wu et al., 2018) trains one global policy network to skip residual blocks. The following works extend this idea to even finer-grained granularity levels, e.g., the filter level (Lin et al., 2017; Chen et al., 2018) or the bit level (Shen et al., 2020; Fu et al., 2020). Different from previous works, our D$^2$-GCN framework is the first attempt at dynamic inference in the context of GCNs. More importantly, D$^2$-GCN leverages the unique structures of GCNs to exploit GCN-specific data-dependent strategies from three different granularities, i.e., **node-wise**, **edge-wise**, and **bit-wise**, to largely boost the accuracy-efficiency trade-offs frontiers of GCNs, and further makes these strategies to facilitate the development of deeper GCNs.

## 3    THE PROPOSED D$^2$-GCN FRAMEWORK

In this section, we first introduce the preliminaries of GCNs and the motivating analysis to support our D$^2$-GCN framework, and then present the detailed design of D$^2$-GCN and its training pipeline.

### 3.1    PRELIMINARIES OF GCNS

**GCN general formulation.** For a given graph $\mathcal{G} = (\mathcal{V}, \mathcal{E})$ with $n$ nodes $v_i \in \mathcal{V}$, $m$ edges $(v_i, v_j) \in \mathcal{E}$, and an adjacent matrix $A \in \mathbb{R}^{N \times N}$ to represent the connectivity information, where the non-zero entries represent the existing connections among different nodes. Also, the node degree for each node $v_i \in \mathcal{V}$ is defined as $d_i = \sum_j A_{ij}$ and the diagonal degree matrix $D$ is formed with $D_{ii} = d_i$. For each layer $l$ of a GCN, the hidden features of the nodes are represented by the feature matrix $x_l \in \mathbb{R}^{N \times H}$, where $H$ denotes the hidden feature dimension of each node. Thus, a GCN layer can be formulated as:

$$x_{l+1} = \text{ACT}_l \left( \hat{A} x_l W_l \right), \tag{1}$$

where $\hat{A}$ is the normalized version of $A$: $\hat{A} = D^{-\frac{1}{2}} A D^{-\frac{1}{2}}$, $\text{ACT}_l$ represents the activation function of layer $l$, and $W_l$ represents the weights of layer $l$. The inference process of one GCN layer can be viewed as two separated phases: *Aggregation* and *Combination*. $\hat{A} x_l$ represents the *Aggregation* phase which aggregates the 1-hop neighbors of each node into a unified feature vector; after that, during the *Combination* phase, $\hat{A} x_l$ is transformed to $\hat{A} x_l W_l$ via a feed-forward layer. Meanwhile, there are also some works that define the combination phase inside the aggregation phase, thus merging the two phases into a single message passing process with the self-loop (Hamilton, 2020).

**Complexity analysis of GCNs.** The computational complexity of GCN inferences can be represented as:

$$\mathcal{O}(LmH + LnH^2), \tag{2}$$

where $L$ is the total number of GCN layers, $n$ is the total number of nodes, $m$ is the total number of edges, and $H$ is hidden feature dimension of each node (Chiang et al., 2019).

## 3.2 D$^2$-GCN: MOTIVATING ANALYSIS

**Causes of GCNs' prohibitive inference cost.** The inefficiency of GCNs mainly comes from two aspects: First, graphs are often very large as exacerbated by their complex and irregular neighbor connections, which lead to a prohibitive amount of nodes and edges, i.e., $m$ and $n$ in Eq. 2, respectively. For example, a graph dataset can come with as many as 169,343 nodes and 39,561,252 edges (Hu et al., 2020), which can cause both large computational and data movement costs. Second, the dimension of GCNs' node feature vectors, i.e., $H$ in Eq. 2, can be very large, e.g., each node in the Citeseer graph has a feature dimension of 3,703, leading to a high workload during the feed-forward computation in the combination step, especially when the computations are conducted with full precision (i.e., 32-bits float point). Accordingly, a straightforward way to reduce the aforementioned costs associated with GCN inference is to reduce the number of (1) nodes, i.e., $n$ in Eq. 2, (2) connections between nodes, i.e., edges, $m$ in Eq. 2, and (3) features of each node, i.e., $H$ in Eq. 2, whereas naively reducing these parameters can hurt GCNs' model capacity and thus their achievable task accuracy.

**Causes of deeper GCNs' training difficulty.** Deeper GCNs are consistently observed to suffer from accuracy drop as compared to their shallower counterparts, regardless of the adopted GCN designs (Kipf & Welling, 2016; Pham et al., 2017; Rahimi et al., 2018; Xu et al., 2018). (Li et al., 2018a; Zhao & Akoglu, 2019) propose that the accuracy drop is resulted from the GCN's over-smoothing issue, i.e., repeatedly applying GCN layers many times will make the hidden features of different nodes converge to similar values. Based on that, some regularizations are proposed to enable a deeper GCN to achieve a higher accuracy, e.g., randomly drop out certain edges of the input graph during each training iteration in (Rong et al., 2020).

**Not all data/(model components) are equally important.** Recent works in both CNNs (Katharopoulos & Fleuret, 2018; Wang et al., 2018) and GCNs (Veličković et al., 2018; Hamilton et al., 2017) show that not all data samples, e.g., input images or graphs, and model components, e.g., specific Convolutional or Graph Convolutional layers, for the same model are equally important for a given task in terms of the achievable accuracy vs. efficiency trade-offs. For example, some of data/(model components) can be skipped without hurting or even boosting the accuracy due to the increased model flexibility. These observations motivate us to consider boosting both GCNs' efficiency and scalability by processing the graphs in a comprehensive data-dependent manner, i.e., for different data, only a fraction of the graph's components, e.g., parts of nodes, edges, or bit-width, are involved into the computations based on the corresponding features of the given graph. The resulting benefits come from two aspects: (1) such a data-dependent design can dynamically allocate more computational budgets to difficult data samples and smaller budgets for simpler data samples to reduce the total inference cost while maintaining the accuracy; and (2) the increased model flexibility resulting from the dynamic inference models can naturally provide a certain regularization effect since different components of a graph may work together or independently in an data-dependent manner to alleviate GCNs' over-smoothing issue discussed in (Li et al., 2020a). As such, deeper GCNs can be more effectively trained thanks to the improved learning capacity enabled by the increased model flexibility.

## 3.3 D$^2$-GCN: SKIPPING STRATEGY DESIGN

**Overview.** Motivated by the analysis in the above subsection, we propose the D$^2$-GCN framework that can dynamically process each graph with a complexity adapting to the data difficulty at three granularities, i.e., node/edge/bit-wise, as shown in Fig. 1. We hypothesize that such a coarse-to-fine strategy can achieve more efficient GCN inference without hurting the accuracy, and maximize the flexibility of GCN structures to provide effective regularizations for training deeper GCNs. Based on the GCN layer formulation in Eq. 1, we can represent each layer of our proposed D$^2$-GCN as:

$$x_{l+1} = g_l^n \odot x_{g_l^b} + (1 - g_l^n) \odot x_l \tag{3}$$

$$x_{g_l^b} = \sum_{k=1}^{K} g_l^{b,k} \odot (\hat{A}_{g_l^e} Q_{B_k}(x_l) Q_{B_k}(w_l)) \tag{4}$$

$$\hat{A}_{g_l^e} = (g_l^e)^T \odot \hat{A} \tag{5}$$

where $x_{l+1}$ and $x_l \in \mathbb{R}^{n \times H}$ denote the feature matrix of layer $l + 1$ and layer $l$, respectively, $w_l \in \mathbb{R}^{H \times H}$ is the weights for the *Combination* phase in layer $l$, $\hat{A} \in \mathbb{R}^{n \times n}$ is the normalized adjacency matrix, and $\odot$ represents the element-wise matrix multiplication (broadcast will be performed if the matrix shapes do not match). All the variables above share the same definition as in Eq. 1. Specifically, in Eq. 3, $g_l^n \in \{0,1\}^{n \times 1}$ represents the output of the gating function for node-wise skipping, and $x_{g_l^b}$ is the feature matrix after enabling the gating function for bit-wise skipping in layer $l$; in Eq. 4, $g_l^b \in \{0,1\}^{n \times K}$

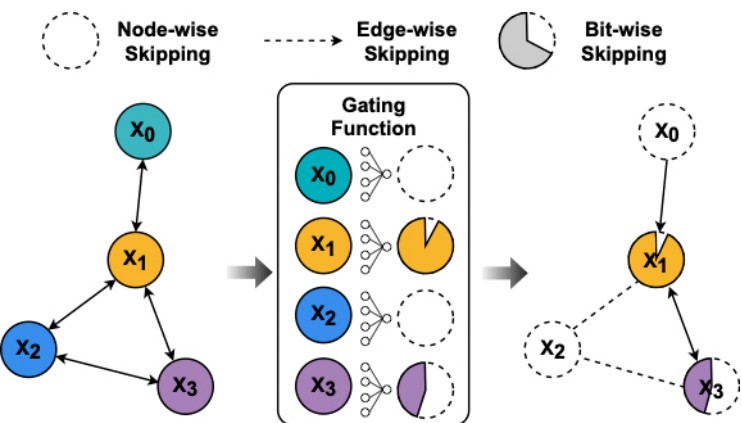

Figure 1: An illustration of our proposed D$^2$-GCN technique in one layer. Specially, $X_1 \sim X_4$ are the 4 nodes in the graph. In this example, the aggregation and combination phases of $X_0$ and $X_2$ are skipped (i.e., **node-wise** skipping as illustrated in Eq. 3) and the hidden features of $X_0$ and $X_2$ are also skipped in the aggregation phases of their neighbors (i.e., **edge-wise** skipping as illustrated in Eq. 5), while the aggregation and combination phases of $X_1$ and $X_3$ are processed with a precision lower than 32-bits float point (i.e., **bit-wise** skipping as illustrated in Eq. 4).

$(g_l^{b,k} \in \{0,1\}^{n \times 1}$ is the $k$-th entry of $g_l^b$) represents the output of the gating function for bit-wise skipping, $Q_{B_k}()$ is the quantization function to quantize weights $w_l$ and feature matrix $x_l$ into $B_k$-bits following the pre-defined quantization bit-width options $B = \{B_1, ..., B_K\}$, and $\hat{A}_{g_l^e}$ is the normalized adjacency matrix after enabling the gating function for edge-wise skipping in layer $l$; and in Eq. 5, $g_l^e \in \{0,1\}^{n \times 1}$ represents the output of the gating function for edge-wise skipping. We elaborate more on how they work in GCNs and how they tackle the causes of GCNs' inference cost below.

**Node-wise skipping (i.e., Eq. 3).** $g_l^n$ makes a binary decision for each node regarding whether to skip its aggregation and combination phases based on its hidden features. In this way, a set of less important nodes identified by $g_l^n$ will not participate the feed-forward computation during each GCN inference and the corresponding hidden features in the feature matrix $x_l$ will be directly passed to the next layer to construct $x_{l+1}$, so that the inference cost (FLOPs, latency, and energy) will be reduced thanks to the smaller $n$ and $m$ in the computational complexity, as analyzed in Eq. 2.

**Bit-wise skipping (i.e., Eq. 4).** $g_l^b$ determines the quantization precision in both the aggregation and combination phases for each node according to its hidden features. Since quantization reduces the feed-forward computation from the most fine-grained bit level, the computational cost for updating each feature can be aggressively reduced, making it feasible to deal with large node feature vectors even on resource constrained platforms.

**Edge-wise skipping (i.e., Eq. 5).** $g_l^e$ determines whether the connection between two nodes will be removed based on the hidden features of the corresponding nodes. Specifically, the connection will be removed when the corresponding item is zero in $\hat{A}_{g_l^e}$, as defined in Eg. 5, and thus result in a smaller inference cost (FLOPs, latency, and energy) thanks to the resulting smaller $m$ in Eq. 2.

**Gating function design.** Inspired by the gating function designs from dynamic CNNs (Wang et al., 2018), we adopt a single feed-forward layer (i.e., fully-connected layer) to map each node's feature (a vector with the shape of $1 \times H$) to the output of the gating functions. For the node-wise and edge-wise skipping, the output of the gating functions is only an element with the shape of $1 \times 1$. For the bit-wise skipping, the output of the gating functions is a vector with the shape of $1 \times K$. $K$ represents the number of bit-width options and we set $K = 4$ in our experiments. Additionally, we follow (Wang et al., 2018) to make the training of the gating functions differentiable via straight through estimators (Bengio et al., 2013). After building the gate functions following the design described above, we surprisingly find that such a simple design can effectively capture the dynamic patterns with very low computational overhead. In our design, it is less than 0.07% of the GCN model's cost in terms of FLOPs, while the gating functions in CNNs can cause an overhead of as large as 12.5% ($\uparrow 179\times$) (Wang et al., 2020).

### 3.4   $\text{D}^2$-GCN: Training Pipeline Design

**Learning objective.** The learning objective of $\text{D}^2$-GCN, $\mathcal{L}_{D^2-GCN}(W, W_G)$, can be formulated as:

$$\mathcal{L}_{D^2-GCN}(W, W_G) = \mathcal{L}_{GCN}(W, W_G) + \alpha\, \mathcal{L}_{comp} \tag{6}$$

where $W = \{w_0, w_1, w_2, ...\}$ is the total set of model weights, $W_G = \{w_{g_0^n}, w_{g_0^e}, w_{g_0^b}, w_{g_1^n}, ...\}$ is the total set of the gating functions' weights, $\mathcal{L}_{GCN}$ is the commonly adopted loss function of graph-based learning tasks like node classification (Kipf & Welling, 2016) or graph classification (Hu et al., 2020), $\alpha$ is a trade-off parameter to balance task performance and efficiency, and $\mathcal{L}_{comp}$ is the computational cost determined by the gating functions. If we use FLOPs as its metric, it can be written as:

$$\mathcal{L}_{comp} = \frac{\sum_{l=1}^{L}(||A_{g_l^e}||_0 H \frac{\sum_{k=1}^{K}||g_l^{b,k}||_0 \frac{B_k}{32}}{n} + ||g_l^n||_0 H^2 \frac{\sum_{k=1}^{K}||g_l^{b,k}||_0 (\frac{B_k}{32})^2}{n})}{\sum_{l=1}^{L}(mH + nH^2)} \tag{7}$$

The variables in Eq. 7 share the same definition as those in Eq. 1, 2, 3, 4, and 5, i.e., $n$ is the total number of nodes, $m$ is the total number of edges, $H$ is the feature dimensions of each node, $A_{g_l^e}$ is the adjacency matrix after enabling the gating function for edge-wise skipping in layer $l$, $g_l^{b,k}$ is the $k$-th entry of $g_l^b$ which is the output of the bit-wise skipping's gating function, and $g_l^n$ represents the output of the gating function for node-wise skipping. It is worth noting that Eq. 7 also indicates how our data-dependant dynamic skipping techniques reduce the inference cost. Specifically, the node-wise skipping will squeeze the cost via reducing the number of nodes ($n$) to $||g_l^n||_0 < n$. The edge-wise skipping will shrink the the number of edges ($m$) to $||A_{g_l^e}||_0 < m$ to reduce the inference cost. The bit-wise skipping will reduce the cost of each matrix multiplication/addition by using a smaller bit-width instead of full precision (32-bits), which can be regarded as further multiplying a factor to the inference cost (i.e., $\frac{\sum_{k=1}^{K}||g_l^{b,k}||_0 \frac{B_k}{32}}{n} < 1$ and $\frac{\sum_{k=1}^{K}||g_l^{b,k}||_0 (\frac{B_k}{32})^2}{n} < 1$)

**Three-stage training pipeline.** Jointly training the GCN and gating functions from scratch could lead to both lower task performance and inferior gating decisions since the gating functions with random initializations in the early training stages may harm the learning process via improper gating strategies. Therefore, we propose a three-stage training pipeline to stabilize the training of $\text{D}^2$-GCN.

*Stage 1:* We pretrain the GCN model with the gating functions fixed and unused. We find this step is indispensable for a decent $\text{D}^2$-GCN design since the gating functions can hardly learn a meaningful gating strategy on top of an under-performed GCN model.

*Stage 2:* We fix the GCN model and train the gating functions only to maximize the task performance by setting the trade-off parameter $\alpha$ in Eq. 6 to be 0. Since randomly initialized gating functions may generate improper gating strategies which may deteriorate the pretrained GCN's performance, this step contributes to generate a decent initialization for the gating functions' weights.

*Stage 3:* We jointly train the GCN model and gating functions based on Eq. 6 to optimize both the task performance and computational cost. After this step, the $\text{D}^2$-GCN trained model is ready to be delivered and deployed onto the target platform.

## 4 EXPERIMENTS RESULTS

In this section, we first introduce the experiment setup and then verify our hypothesis that (1) D$^2$-GCN can boost the efficiency, i.e., achieving better accuracy vs. efficiency trade-offs than both vanilla models and models with SOTA efficient GCN techniques; and (2) D$^2$-GCN can boost the scalability, i.e., achieving a higher accuracy when being applied to deeper GCNs than both vanilla models and models with SOTA scalable GCN techniques. Finally, we perform ablation study on the effectiveness of skipping at different granularities, i.e., how the node-wise, edge-wise, and bit-wise skipping individually affect the achieved accuracy vs. efficiency trade-offs, and visualize the skipping patterns to provide more insights.

### 4.1 EXPERIMENTS SETUP

**Model, datasets, and training hyperparameters.** We evaluate our proposed D$^2$-GCN over **6 GCN model and dataset pairs** (i.e., 3 GCNII (Chen et al., 2020) models on Cora, Citeseer, and Pubmed (Sen et al., 2008) and 3 DeeperGCN (Li et al., 2020a) models on ogbg-molhiv, ogbn-arxiv, and ogbn-proteins (Hu et al., 2020)), as summarized in Table 1. All the training hyperparameters follow the original implementation in (Chen et al., 2020; Li et al., 2020a), and we sweep $\alpha$, which is described in Eq. 6, ranging from 0.1 to 10 in all experiments, and set $K = 4, B_1 = 2, B_2 = 4, B_3 = 8, B_4 = 16$, which are defined in Eq. 4.

**Baselines and evaluation metrics.** We compare our proposed D$^2$-GCN over two types baselines to verify its efficiency and scalability (to deeper GCNs), respectively.

*Efficiency baselines:* Degree-Quant (Tailor et al., 2021), QAT (Fan et al., 2021), and UGS (Chen et al., 2021) are included into the benchmark to verify that our proposed D$^2$-GCN can boost the efficiency in terms of accuracy vs. FLOPs/latency/energy trade-offs.

*Scalability baselines:* We compare our proposed D$^2$-GCN with DropEdge (Rong et al., 2020) and GCNII (Chen et al., 2020), which are the SOTA method to improve the scalability of GCN, aiming at achieving better accuracy with more number of layers.

Specifically, the accuracy is reported as the average over 10 runs with different random seeds, the FLOPs metric follows the computation complexity analyzed in (Chiang et al., 2019) and the adaptation to quantized models in (Fu et al., 2020), and latency/energy is simulated using the architecture described in (Qin et al., 2020).

### 4.2 D$^2$-GCN BOOSTS THE EFFICIENCY

By comparing our D$^2$-GCN with SOTA efficient GCN techniques on different models and datasets in Table 2, we can observe the consistent effectiveness of D$^2$-GCN in boosting the efficiency of GCNs. Specifically, our proposed D$^2$-GCN consistently surpasses all the competitors by achieving $\downarrow 1.1\times \sim \downarrow 37.0\times$ FLOPs/$\downarrow 1.6\times \sim \downarrow 8.4\times$ latency/$\downarrow 1.6\times \sim \downarrow 8.4\times$ energy reduction, while achieving a comparable or even higher accuracy ($\downarrow 0.5\% \sim \uparrow 5.6\%$). We conjecture that this is because (1) UGS (Chen et al., 2021) only considers the redundancy from edge-wise and node-wise while Degree-Quant (Tailor et al., 2021) or QAT (Fan et al., 2021) only considers the redundancy from bit-wise; In contrast, our proposed **D$^2$-GCN simultaneously squeezes out unnecessary costs of GCNs from the edge-wise, node-wise, and bit-wise level**; and (2) UGS (Chen et al., 2021), Degree-Quant (Tailor et al., 2021), and QAT (Fan et al., 2021) do not leverage the assumption that not all data/model components are equally important, which has been verified by (Wang et al., 2020; Katharopoulos & Fleuret, 2018), thus they may only achieve sub-optimal accuracy vs. efficiency trade-offs because hard/easy samples are assigned insufficient/redundant computational resource,

Table 1: The statistics of the GCN models and datasets.

| Model | #Layers | Dataset | Dataset Type | #Graphs | Avg. #Nodes | Avg. #Edges |
|---|---|---|---|---|---|---|
| GCNII (Chen et al., 2020) | 64 | Cora (Sen et al., 2008) | Node classification | 1 | 2,708 | 5,429 |
| | 32 | Citeseer (Sen et al., 2008) | | 1 | 3,327 | 4,732 |
| | 16 | Pubmed (Sen et al., 2008) | | 1 | 19,717 | 44,338 |
| DeeperGCN (Li et al., 2020a) | 112 | ogbn-proteins (Hu et al., 2020) | Node classification | 1 | 132,534 | 39,561,252 |
| | 28 | ogbn-arxiv (Hu et al., 2020) | | 1 | 169,343 | 1,166,243 |
| | 7 | ogbg-molhiv (Hu et al., 2020) | Graph classification | 41,127 | 25.5 | 27.5 |

Table 2: Comparison between $D^2$-GCN and SOTA efficient GCN techniques in terms of accuracy vs. efficiency trade-offs, where vanilla is the original setting of the corresponding models and datasets without any efficient GCN techniques.

| Method | Accuracy (%) | FLOPs (G) | Normalized | |
|---|---|---|---|---|
| | | | Latency (%) | Energy (%) |
| **GCNII@Cora** | | | | |
| Vanilla | 85.5 | 0.74 | 100.0 | 100.0 |
| UGS | 82.4 | 0.59 | 99.9 | 99.9 |
| QAT | 85.1 | 0.06 | 20.0 | 20.0 |
| Degree-Quant | 81.0 | 0.12 | 20.0 | 20.0 |
| **$D^2$-GCN** | **85.6** | **0.02** | **12.6** | **12.2** |
| **$D^2$-GCN Improv.** | ↑0.1 ∼ ↑4.6 | ↓3.0× ∼ ↓37.0× | ↓1.6× ∼ ↓7.9× | ↓1.6× ∼ ↓8.2× |
| **GCNII@Citeseer** | | | | |
| Vanilla | 73.4 | 7.05 | 100.0 | 100.0 |
| UGS | 72.1 | 5.65 | 99.9 | 99.8 |
| QAT | 73.4 | 0.48 | 22.0 | 22.0 |
| Degree-Quant | 69.5 | 1.13 | 22.0 | 22.0 |
| **$D^2$-GCN** | **73.4** | **0.22** | **13.3** | **13.3** |
| **$D^2$-GCN Improv.** | ↑0.0 ∼ ↑3.9 | ↓2.2× ∼ ↓32.0× | ↓1.7× ∼ ↓7.5× | ↓1.7× ∼ ↓7.5× |
| **GCNII@Pubmed** | | | | |
| Vanilla | 80.2 | 20.86 | 100.0 | 100.0 |
| UGS | 78.8 | 16.71 | 99.9 | 99.9 |
| QAT | 79.5 | 1.34 | 24.7 | 24.7 |
| Degree-Quant | 77.8 | 3.29 | 24.7 | 24.7 |
| **$D^2$-GCN** | **79.7** | **1.13** | **14.9** | **14.8** |
| **$D^2$-GCN Improv.** | ↓0.5 ∼ ↑1.7 | ↓1.2× ∼ ↓18.5× | ↓1.7× ∼ ↓6.7× | ↓1.7× ∼ ↓6.8× |
| **DeeperGCN@ogbg-molhiv** | | | | |
| Vanilla | 78.6 | 483.14 | 100.0 | 100.0 |
| UGS | 75.8 | 386.81 | 97.1 | 99.9 |
| QAT | 72.9 | 30.58 | 21.1 | 21.1 |
| Degree-Quant | 78.2 | 75.83 | 21.1 | 21.1 |
| **$D^2$-GCN** | **78.5** | **28.73** | **12.3** | **12.4** |
| **$D^2$-GCN Improv.** | ↓0.1 ∼ ↑5.6 | ↓1.1× ∼ ↓16.8× | ↓1.7× ∼ ↓8.1× | ↓1.7× ∼ ↓8.1× |
| **DeeperGCN@ogbn-arxiv** | | | | |
| Vanilla | 71.9 | 81.87 | 100.0 | 100.0 |
| UGS | 71.4 | 66.12 | 98.2 | 98.2 |
| **$D^2$-GCN** | **72.2** | **12.37** | **14.7** | **14.5** |
| **$D^2$-GCN Improv.** | ↑0.3 ∼ ↑0.8 | ↓5.3× ∼ ↓6.6× | ↓6.7× ∼ ↓6.8× | ↓6.8× ∼ ↓6.9× |
| **DeeperGCN@ogbn-proteins** | | | | |
| Vanilla | 85.8 | 405.17 | 100.0 | 100.0 |
| UGS | 85.2 | 366.67 | 97.7 | 97.8 |
| Degree-Quant | 85.1 | 111.16 | 23.2 | 23.4 |
| **$D^2$-GCN** | **85.3** | **72.24** | **11.9** | **11.9** |
| **$D^2$-GCN Improv.** | ↓0.5% ∼ ↑0.2% | ↓1.5× ∼ ↓5.6× | ↓1.9× ∼ ↓8.4× | ↓2.0× ∼ ↓8.4× |

whereas our proposed **$D^2$-GCN is a data-dependent framework** that processes each data sample with an adaptive model capacity/cost, maximizing the accuracy vs. efficiency trade-offs.

### 4.3 $D^2$-GCN Boosts the Scalability

As summarized in Table 3, by varying the number of layers from 4 to 64 on different datasets, our proposed $D^2$-GCN (1) **achieves a higher accuracy as we increase the number of layers** (i.e., ↑0.8% ∼ ↑5.1% accuracy boost) and (2) is **on par with SOTA techniques for a better GCN scalability** (i.e., GCNII (Chen et al., 2020) and DropEdge (Rong et al., 2020)) with a comparable or even better accuracy under the same number of layers, especially on deeper models (↓0.4% ∼ ↑38.6% for models with 64 layers). This set of experiments indicates that with the proposed node-wise, edge-wise, and bit-wise skipping at different granularities, our

Table 3: Comparison between $D^2$-GCN and SOTA scalable GCN techniques, where vanilla is the original setting used in (Kipf & Welling, 2016) without any scalable GCN techniques, which achieves a lower accuracy when going deeper.

| Dataset | Method | Accuracy (%) with Num. layers | | |
|---|---|---|---|---|
| | | 4 | 16 | 64 |
| Cora | Vanilla | 80.4 | 64.9 | 28.7 |
| | DropEdge | 82.0 | 75.7 | 49.5 |
| | GCNII | 82.6 | 84.2 | 85.5 |
| | **$D^2$-GCN** | **80.5** | **84.2** | **85.6** |
| | **$D^2$-GCN Improv.** | ↓2.1 ∼ ↑0.1 | ↑0.0 ∼ ↑19.3 | ↑0.1 ∼ ↑56.9 |
| Citeseer | Vanilla | 67.6 | 18.3 | 20.0 |
| | DropEdge | 70.6 | 57.2 | 34.4 |
| | GCNII | 68.9 | 72.9 | 73.4 |
| | **$D^2$-GCN** | **69.7** | **73.1** | **73.0** |
| | **$D^2$-GCN Improv.** | ↓0.9 ∼ ↑2.1 | ↑0.2 ∼ ↑54.8 | ↓0.4 ∼ ↑53.0 |
| Pubmed | Vanilla | 76.5 | 40.9 | 35.3 |
| | DropEdge | 79.4 | 78.5 | 61.5 |
| | GCNII | 78.8 | 80.2 | 79.7 |
| | **$D^2$-GCN** | **79.0** | **79.5** | **79.8** |
| | **$D^2$-GCN Improv.** | ↓0.4 ∼ ↑2.5 | ↓0.7 ∼ ↑38.6 | ↑0.1 ∼ ↑44.5 |

proposed $D^2$-GCN can alleviate the over-smoothing issue, as discussed in (Li et al., 2018a), and thus facilitate GCNs to achieve a higher accuracy when going deeper. This matches our hypothesis that such data-dependent **skipping techniques can enforce a regularization** effect since different components of a graph may work together or independently in an data-dependent manner to facilitate the removal of the over-smoothing issue.

## 4.4 ABLATION STUDY: EFFECTIVENESS OF SKIPPING AT DIFFERENT GRANULARITIES

As introduced in Figure 1, $D^2$-GCN integrates node-wise, edge-wise, and bit-wise skipping, which target different granularities. For a better understanding of the proposed $D^2$-GCN, we conduct an ablation study on the effectiveness of each skipping technique on both node classification (Cora (Sen et al., 2008)) and graph classification datasets (ogbn-molhiv (Hu et al., 2020)), i.e., disabling one skipping

Table 4: Comparison among node-wise, edge-wise, and bit-wise skipping at different granularities in the proposed $D^2$-GCN, where vanilla is the original setting of the corresponding models and datasets without any efficient GCN techniques.

| Method | Accuracy (%) | FLOPs (G) | Normalized | |
| --- | --- | --- | --- | --- |
| | | | Latency (%) | Energy (%) |
| **GCNII@Cora** | | | | |
| Vanilla | 85.5 | 0.74 | 100.0 | 100.0 |
| $D^2$-GCN w/o node-wise skipping | 85.6 (↑0.1) | 0.03 (↓24.7×) | 13.6 (↓7.4×) | 13.7 (↓7.3×) |
| $D^2$-GCN w/o edge-wise skipping | 85.6 (↑0.1) | 0.03 (↓24.7×) | 13.6 (↓7.4×) | 13.6 (↓7.4×) |
| $D^2$-GCN w/o bit-wise skipping | 85.5 (↑0.0) | 0.67 (↓1.1×) | 88.2 (↓1.1×) | 88.5 (↓1.1×) |
| **$D^2$-GCN** | **85.6 (↑0.1)** | **0.02 (↓37.0×)** | **12.6 (↓7.9×)** | **12.2 (↓8.2×)** |
| **DeeperGCN@ogbg-molhiv** | | | | |
| Vanilla | 78.6 | 484.14 | 100.0 | 100.0 |
| $D^2$-GCN w/o node-wise skipping | 75.5 (↓3.1) | 40.87 (↓11.8×) | 14.2 (↓7.0×) | 14.2 (↓7.0×) |
| $D^2$-GCN w/o edge-wise skipping | 77.7 (↓0.9) | 28.91 (↓16.7×) | 13.0 (↓7.7×) | 13.2 (↓7.6×) |
| $D^2$-GCN w/o bit-wise skipping | 77.5 (↓1.1) | 338.2 (↓1.4×) | 91.0 (↓1.1×) | 91.5 (↓1.1×) |
| **$D^2$-GCN** | **78.5 (↓0.1)** | **28.7 (↓16.9×)** | **12.3 (↓8.1×)** | **12.4 (↓8.1×)** |

technique of our proposed $D^2$-GCN each time, and summarize the results in Table 4. We can make the following observations: (1) only incorporating **one skipping technique corresponding to one granularity can still reduce the inference cost (e.g., FLOPs, energy, and latency) of GCN models but leading to an obviously lower accuracy** (↓1.1× ∼ ↓24.7× less FLOPs/↓1.1× ∼ ↓7.7× less latency/↓1.1× ∼ ↓7.6× less energy, but ↓3.1% ∼ ↑0.1% accuracy decrease) than the baselines; and (2) $D^2$-GCN **integrating node-wise, edge-wise, and bit-wise skipping simultaneously can achieve better accuracy vs. efficiency trade-offs** than its vanilla baselines disabling one/two of the three (↓1.5× ∼ ↓33.5× less FLOPs/↓1.1× ∼ 7.4× less latency/1.1× ∼ 7.4× less energy, and ↑0.0% ∼ ↑3.0% higher accuracy), validating the importance of skipping at all the three different granularities, as described in Section 3.3.

## 4.5 MORE INSIGHTS FROM VISUALIZATION OF THE SKIPPING PATTERNS

To better understand and illustrate why $D^2$-GCN surpasses all the efficient GCN competitors in terms of accuracy vs. efficienct trade-offs, as demonstrated in Table 2, we visualize the layer-wise FLOPs

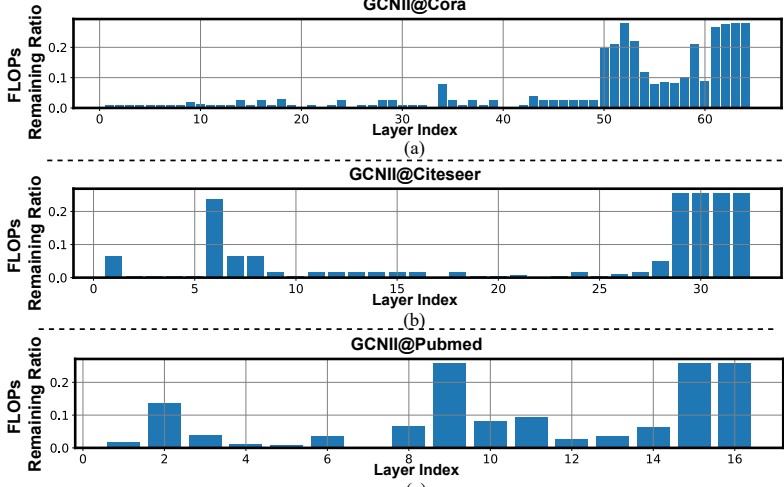

Figure 2: Visualization of the FLOPs remaining ratio (i.e., the ratio between FLOPs of GCNs w/ and w/o our proposed $D^2$-GCN) of different layers when our proposed $D^2$-GCN is added on top of a GCNII model with the (a) Cora, (b) Citeseer, and (c) Pubmed datasets.

remaining ratio (the lower means the more FLOPs reduction is enabled by our proposed $D^2$-GCN) in Figure 2. We can extract the following insights: (1) the imbalance FLOPs remaining ratios among different layers verify that the hypothesis, **not all data/model components are equally important, also holds for GCNs**; (2) such imbalance FLOPs remaining ratios play an important role in enabling the proposed $D^2$-GCN to achieve SOTA accuracy vs. efficient trade-offs by **dynamically allocating more computational budgets to difficult data samples and/or important model components and lower budgets for simple data samples and/or unnecessary model components** to reduce the total inference cost while maintaining the accuracy, because the SOTA efficient GCN baseline techniques in Section 4.2 are using either the same node and edge relationships (i.e. adjacency matrix $A$ in Eq. 1) (Tailor et al., 2021; Fan et al., 2021) or the same bit-widths (Chen et al., 2021) along all layers; and (3) **the latter layers (i.e., closer to the final classifier) tend to maintain higher FLOPs remaining ratios**, i.e., they are less redundant and more important than the former layers, which can provides insights for more efficient GCN architecture design.

## 5 CONCLUSION

In this paper, we propose a Data-Dependent GCN framework, $D^2$-GCN, which is the first dynamic inference framework dedicated to GCNs. Specifically, three types of low-cost gating functions are integrated to realize data-dependent dynamic skipping at multiple granularities: node-wise, edge-wise, and bit-wise, targeting fewer nodes to combine, fewer edges to aggregate, and lower bit-precision to compute. Besides achieving SOTA accuracy vs. efficiency trade-offs to facilitate the deployment of GCNs onto resource-constrained applications, our proposed $D^2$-GCN can also help on alleviating the over-smoothing issue of GCNs to improve the accuracy of GCNs when they go deeper, thus removing the barriers of handling real-world large graphs and unleashing the potential of more sophisticated GCN architectures. As such, $D^2$-GCN opens up a new knob to not only boost GCNs' inference efficiency but also help to build deeper and more powerful GCNs in the data-dependant manner.

## 6 REPRODUCIBILITY STATEMENT

Regarding our efforts that have been made to ensure reproducibility, we provide the implementation details in Appendix A.

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

## A    IMPLEMENTATION DETAILS

We provide the implementation details of our experiments here, including our $D^2$-GCN (1) on Cora, Citeseer and Pubmed datasets (Sen et al., 2008) on top of GCNII models (Chen et al., 2020) and (2) ogbg-molhiv, obgn-arXiv, and ogbn-proteins dataset (Hu et al., 2020) on top of DeeperGCN models (Li et al., 2020a) in Section 4.2 $\sim$ 4.5.

**$D^2$-GCN models on Cora, Citeseer and Pubmed datasets**.  All the reported $D^2$-GCN models on these 3 datasets on top of GCNII models (Chen et al., 2020) follow the same training recipe (including data pre-processing) with the one proposed in (Chen et al., 2020). For Cora dataset, the number of layers and hidden dimensions are both 64, the dropout rate is 0.6, the weight decay of graph convolutional layers is 0.01, and the weight decay of the fully-connected layers at the beginning and the end is 5e-4. For Citeseer dataset, the number of layer is 32, the hidden dimensions is 256, the dropout rate is 0.7, and the weight decay setting is the same with the one for Cora dataset. For Pubmed dataset, the number of layers is 16, hidden dimension is 256, the dropout rate is 0.7, the weight decay of all layers is 5e-4. Note that, the $\alpha$ described in Eq. 6 sweeps from [0.1, 0.2, 0.5, 1.0, 2.0, 5.0, 10.0], and the best $\alpha$ for Cora, Citeseer, and Pubmed datasets are 10, 0.2, and 1.0, respectively.

**$D^2$-GCN models on ogbg-molhiv, ogbn-arxiv, and ogbn-proteins dataset.**  All the reported $D^2$-GCN models on these 3 datasets on top of DeeperGCN models (Li et al., 2020a) follow the same training recipe (including data pre-processing) as the one proposed in (Li et al., 2020a). For ogbg-molhiv dataset, the model (1) is built with the block type as res+, the normalizatio type as BatchNorm, the number of layers as 7, the hidden dimension as 256, and the graph convolution aggregation type is $softmax$; (2) is trained with the learning rate as 0.0001 and the dropout rate as 0.2. For ogbn-arxiv dataset, the model (1) shares the same architecture configurations as the one for ogbg-molhiv dataset except stacking for 28 layers instead of 7 layers, decreasing the hidden dimensions from 256 to 128, and using $softmax_{sg}$ as the graph convolution aggregation type; (2) is trained with the learning rate as 0.001 and the dropout rate as 0.5. For ogbn-proteins dataset, the model (1) is built with the block type as res+, the normalizatio type as LayerNorm, the number of layers as 112, the hidden dimension as 64, and the graph convolution aggregation type is $softmax$; (2) is trained with the learning rate as 0.001 and the dropout rate as 0.1. Same with the setting in $D^2$-GCN models on Cora, Citeseer and Pubmed datasets, the $\alpha$ described in Eq. 6 sweeps from [0.1, 0.2, 0.5, 1.0, 2.0, 5.0, 10.0], and the best $\alpha$ for ogbg-molhiv, ogbn-arxiv, and ogbn-proteins datasets are 1.0, 1.0, and 10.0, respectively.

