# OpenReview forum: "D$^2$-GCN: Data-Dependent GCNs for Boosting Both Efficiency and Scalability"
_ICLR.cc/2022/Conference — ICLR 2022 Submitted_

### Official Review · Reviewer_AVzB · 2021-11-02

**Correctness:** 3
**Technical Novelty And Significance:** 3
**Empirical Novelty And Significance:** 3
**Recommendation:** 3
**Confidence:** 3

**Main Review:**

Strengths:

(1) This paper is well motivated.

(2) The Introduction and Related works parts are well written.

(3) Experiments on top of various SOTA GCNs and datasets validate the effectiveness and advantages of D$^2$-GCN.


Weaknesses:

(1) The paper has limited technique contributions. The problem of boosting the efficiency and scalability of GCNs has been fully studied by existing works. The ideas of *Node-wise skipping* and *Edge-wise skipping* are not novel enough. Please refer to the following references:

- [KDD 2019] [Cluster-GCN] An Efficient Algorithm for Training Deep and Large Graph Convolutional Networks
- [ICCV 2019] [DeepGCNs] Can GCNs Go as Deep as CNNs
- [ICLR 2016] [GGS-NNs] Gated Graph Sequence Neural Networks
- [ICLR 2020] [DropEdge] Towards Deep Graph Convolutional Networks on Node Classification
- [NIPS 2020] [DropNode] Graph Random Neural Networks for Semi-Supervised Learning on Graphs

(2) The main contribution of this work is reducing the GCN inference cost. The data-dependent dynamic skipping strategies achieve this. I suggest the authors add theoretical complexity analysis comparison to better support this.

(3) The skipping strategy are confusing:
- the node-wise skipping parameter $G_l^n$ is 0 or 1. However, the authors claim $G_l^n \in R^{N*1}$. The problems also exist for $G_l^b$ and $G_l^e$.
- The gate function design part is important. However, the description is too brief.

(4) In Eq.(6), $L_{comp}$ denotes the computational cost determined by the decisions made by the gating functions. The paper lacks a detailed calculation method.

(5) This paper lacks hyperparameters experiments about trade-off parameter $\alpha$, which can intuitively provide the trade-off between accuracy and efficiency.

(6) The writing needs improvements. There are many typos and problems with the mathematical symbols, sentence structure, such as

- The real number (M,N), vector ($G_l^n$) and matrix ($X$) are denoted as uppercase characters.
- The sentence in the first paragraph of page 6 is too long and hard to read.
- "Specially, where $X_1$ ∼ $X_4$ are the 4 nodes in the graph."?
- "input images or graphes"?


**Summary Of The Paper:**

This paper proposes a Data-Dependent GCN framework (D$^2$-GCN) that integrates data-dependent dynamic skipping at multiple granularities. D$^2$-GCN is achieved by identifying the importance of node features via a low-cost indicator and thus is simple and generally applicable to various graph-based learning tasks. Experiments certify the effectiveness and efficiency of D$^2$-GCN.

**Summary Of The Review:**

Overall, this work is well motivated. However, the theoretical complexity analysis is missing and the methodology part remains confused. The technique contributions are limited.

---

> ### Author Response · Authors · 2021-11-23
> **Response to Reviewer AVzB**
>
> > + Q1: The paper has limited technique contributions. The problem of boosting the efficiency and scalability of GCNs has been fully studied by existing works. The ideas of Node-wise skipping and Edge-wise skipping are not novel enough.
>
> A1: We agree that there exist some prior works to boost the efficiency and scalability of GCNs, but 1) **our work is orthogonal to them** (e.g., GCNII [1] and DeeperGCN [2] are proposed to boost the scalability of GCNs, and our proposed D$^2$-GCN can be added on top of them to achieve better efficiency and scalability); 2) we focus on **boosting both the accuracy vs. efficiency trade-off and the trainability of deeper GCNs**, while previous gating or dropping techniques only focus on the latter, e.g., [3] uses RNN and LSTM as the gating function without considering the overhead of those heavy gating functions may be larger than the GCN’s inference cost. Similarly, [4] is only applicable to training period with no inference cost reduction; 3) we **squeeze the redundancy of GCNs from multiple granularities**: node-wise, edge-wise, and bit-wise, while previous works only focus on node-wise [3] or edge-wise [4] only.
>
> > + Q2: The main contribution of this work is reducing the GCN inference cost. The data-dependent dynamic skipping strategies achieve this. I suggest the authors add theoretical complexity analysis comparison to better support this.
>
> A2: Thank you for your suggestions! We have added a detailed formula of the computational cost in Eq. 7 in the updated manuscript. In a nutshell, the computational complexity of GCN inference can be represented as: $\mathcal{O}(LmH + LnH^2)$ (Eq. 2 in the updated manuscript). Based on that, the node-wise skipping will squeeze the cost via reducing the number of nodes ($n$). The edge-wise skipping will further squeeze the cost via a smaller number of edges ($m$) in the graph. The bit-wise skipping will reduce the cost of each matrix multiplication/addition by using a smaller bit-width instead of full precision (32-bits), which can be regarded as multiplying a factor (<1) to $LmH$ and $LnH^2$. A more detailed version of the theoretical complexity analysis has been added in Section 3.4 of the updated manuscript.
>
> > + Q3: The skipping strategy are confusing: The node-wise skipping parameter Gnl is 0 or 1. However, the authors claim Gnl ∈ RN*1. The problems also exist for Gbl and Gel
> The gate function design part is important. However, the description is too brief.
>
> A3: Thank you for pointing it out. Regarding the node-wise, edge-wise, and bit-wise skipping parameters, we have corrected them in the updated manuscript. Regarding the gating function design, we adopt a single feed-forward layer (i.e., fully-connected layer) to map each node’s feature (a vector with the shape of $1 \times H$) to the output of the gating functions. For the node-wise and edge-wise skipping, the output of the gating functions is only an element with the shape of $1 \times 1$. For the bit-wise skipping, the output of the gating functions is a vector with the shape of $1 \times K$. $K$ represents the number of bit-width options and we set $K=4$ in our experiments. We have also added that information in the updated manuscript.
>
> > + Q4: In Eq.(6), $\mathcal{L}_{comp}$ denotes the computational cost determined by the decisions made by the gating functions. The paper lacks a detailed calculation method.
>
> A4: Thank you for your suggestions! We have added the detailed calculation method of the computational cost in Eq. 7 in the updated manuscript.

---

> > ### Author Response · Authors · 2021-11-23
> > **Response to Reviewer AVzB**
> >
> > > + Q5: This paper lacks hyperparameters experiments about trade-off parameter α, which can intuitively provide the trade-off between accuracy and efficiency.
> >
> > A5: Thank you for your suggestion! We provide the hyperparameters experiments about the trade-off parameter α in the table below. If the trade-off parameter α is too large, it will reduce the inference FLOPs a lot (e.g., ↓61.7x in GCNII@Cora), but also obviously degrades the accuracy (e.g., -3.8% in GCNII@Cora). However, if it is too small, it can achieve a comparable accuracy with the baseline (e.g., -0.1% in DeeperGCN@ogbg_molhiv) but does not hit the optimal accuracy vs. efficiency trade-off as compared to other trade-off parameters. Specifically, setting the parameter to be 0.2 and 0.1 on DeeperGCN@ogbg_molhiv both achieve 78.5% accuracy, but setting it to be 0.2 saves more inference FLOPs (↓16.8x vs. ↓13.1x). So there exists a sweet point for the trade-off parameter α, and we sweep it from a pre-set list in our experiments, as mentioned in Appendix A.
> >
> > | Model@Dataset | Trade-off parameter | Accuracy (%) | FLOPs (G)|
> > |:-:|:-:|:-:|:-:|
> > | | Baseline | 85.5 | 0.740 |
> > | | 1.0 | 81.7 (-3.8) | 0.012 (↓61.7x) |
> > | GCNII@Cora | 0.5 | 83.5 (-2.0) | 0.017 (↓43.5x) |
> > | | 0.2 | 85.2 (-0.3) | 0.021 (↓35.2x) |
> > | | 0.1 | 85.6 (+0.1) | 0.024 (↓30.8x) |
> > | - | - | - | - |
> > | | Baseline | 78.6 | 483.14 |
> > | | 1.0 | 73.1 (-5.5) | 12.32 (↓39.2x) |
> > | DeeperGCN@ogbg_molhiv | 0.5 | 74.2 (-4.4) | 20.53 (↓23.5x) |
> > | | 0.2 | 78.5 (-0.1) | 28.73 (↓16.8x) |
> > | | 0.1 | 78.5 (-0.1) | 36.94 (↓13.1x) |
> >
> > > + Q6: The writing needs improvements. There are many typos and problems with the mathematical symbols, sentence structure.
> >
> > A6: Thank you for pointing them out! We have corrected them in our updated manuscript and will perform more careful proofreading for the final version.
> >
> > *[1] Chen, Ming, et al. "Simple and deep graph convolutional networks." International Conference on Machine Learning. PMLR, 2020.*
> >
> > *[2] Li, Guohao, et al. "Deepergcn: All you need to train deeper gcns." arXiv preprint arXiv:2006.07739 (2020).*
> >
> > *[3] Li Y, Tarlow D, Brockschmidt M, Zemel R. Gated graph sequence neural networks. arXiv preprint arXiv:1511.05493. 2015 Nov 17.’*
> >
> > *[4] Rong, Yu, et al. "Dropedge: Towards deep graph convolutional networks on node classification." arXiv preprint arXiv:1907.10903 (2019)*.

---

> > ### Comment · Reviewer_AVzB · 2021-11-23
> > **Some responses are not very convicing**
> >
> > Thanks for responding to my concerns, but I am still unsatisfactory.
> >
> > Specifically, the authors claim that: "we focus on boosting both the accuracy vs. efficiency trade-off and the trainability of deeper GCNs". However, this argument may not stand. Regarding the accuracy, as shown in Table 2, the proposed method has very limited improvement. It even performs worse in several cases.
> >
> > By the way, did the authors use the boldface to mark the best results or the results of the proposed method? This should be noted in the table caption.

---

> > > ### Author Response · Authors · 2021-11-23
> > > **Additional Response to Reviewer AVzB**
> > >
> > > Thank you for your quick response! We answer the additional questions regarding your latest reply below.
> > >
> > > > Q1: Specifically, the authors claim that: "we focus on boosting both the accuracy vs. efficiency trade-off and the trainability of deeper GCNs". However, this argument may not stand. Regarding the accuracy, as shown in Table 2, the proposed method has very limited improvement. It even performs worse in several cases.
> > >
> > > A1: We hope to humbly remind that your mentioned cases of “even performs worse in several cases” in Table 2 correspond to the comparison with the vanilla settings (i.e., the original setting of the corresponding models and datasets), which **come with much more FLOPs (i.e., 5.6x ~ 37.0x more FLOPs) than our proposed D$^2$-GCN**. To better illustrate our improvements as compared to SOTA efficient GCN techniques in terms of accuracy vs. efficiency trade-offs, we summarize them in the table below excluding the vanilla setting for avoiding potential confusions. From this table, we can see that our proposed D$^2$-GCN can outperform SOTA efficient GCN techniques with a consistently higher accuracy (i.e., ↑0.0 ~ ↑5.6) and lower required FLOPs (i.e., ↓1.1× ~ ↓29.5× ).
> > >
> > > | | Accuracy (%) | FLOPs (G)|
> > > |:-:|:-:|:-:|
> > > | |GCNII@Cora | |
> > > | D$^2$-GCN Improv. | ↑0.5 ~ ↑4.6 | ↓3.0× ~ ↓29.5×|
> > > | | GCNII@Citerseer | |
> > > | D$^2$-GCN Improv. | ↑0.0 ~ ↑3.9 | ↓2.2× ~ ↓25.7×|
> > > | | GCNII@Pubmed | |
> > > | D$^2$-GCN Improv. | ↑0.2 ~ ↑1.7 | ↓1.2× ~ ↓14.8×|
> > > | | GCNII@ogbg-molhiv| |
> > > | D$^2$-GCN Improv. | ↑0.3 ~ ↑5.6 | ↓1.1× ~ ↓13.5×|
> > > | | GCNII@ogbn-arxiv | |
> > > | D$^2$-GCN Improv. | ↑0.8 ~ ↑0.8 | ↓5.3× ~ ↓5.3× |
> > > | | GCNII@ogbn-proteins | |
> > > | D$^2$-GCN Improv. | ↑0.1 ~ ↑0.2 | ↓1.5× ~ ↓5.1× |
> > >
> > > Meanwhile, we hope to further humbly emphasize that our proposed **D$^2$-GCN not only boosts the accuracy vs. efficiency trade-off but also the trainability of deeper GCNs**. As the latter is **difficult to observe from merely the accuracy improvement and thus leads to your opinion of “very limited improvement”**. Hence, to better illustrate our D$^2$-GCN’s effectiveness in boosting the trainability of deeper GCNs, we follow DropEdge [1] to use the distance of different intermediate layers’ outputs as the metric for measuring the over-smoothing level, under which a larger distance means the over-smoothing issue is less serious, thus corresponding to a better trainability. Based on the results summarized in the table below (our proposed D$^2$-GCN are highlighted in bold), we can observe that our proposed D$^2$-GCN outperforms GCNII [4] in terms of capability to alleviate the over-smoothing issue, as it consistently  enlarges the distance of different intermediate layers by 1.38x ~ 2.40x.
> > >
> > > |Method | Model@Dataset | Accuracy (%) | Avg. Distance After Training|
> > > |:-:|:-:|:-:|:-:|
> > > |Baseline | GCNII@Cora | 85.5 | 1.59 |
> > > | **Ours** | GCNII@Cora | **85.6 (+0.1)** | **2.19 (1.38x)** |
> > > | Baseline | GCNII@Citeseer | 73.4 | 1.27 |
> > > | **Ours** | GCNII@Citeseer | **73.4 (+0.0)** | **3.05 (2.40x)** |
> > >
> > > > Q2: By the way, did the authors use the boldface to mark the best results or the results of the proposed method? This should be noted in the table caption.
> > >
> > > A2: Thank you for your suggestions! The boldface is used to mark the proposed method. We have also added this information to the table caption of our latest manuscript.

---

> > > > ### Author Response · Authors · 2021-11-23
> > > > **Additional Response to Reviewer AVzB**
> > > >
> > > > Besides your review comments, we would also like to kindly draw your attention to some other clarification or updates we have made for our proposed D$^2$-GCN during rebuttal.
> > > >
> > > > 1) Technically, our work contributes a new systematic way to remove the redundancy of GCNs from all/various dimensions with a low overhead, leading to large savings in GCN inference.
> > > >
> > > > 2) We provide a brief theoretical analysis below to explain why our D$^2$-GCN can be regarded as a special implicit regularization to alleviate GCNs’ over-smoothing issue.
> > > >
> > > >
> > > > *As mentioned by DropEdge [1], the over-smoothing issue can be measured by the distance of different intermediate layers’ outputs [1]. A larger distance means that the over-smoothing issue is less serious. As proved in Theorem 1 of [3], the upper bound of such a distance will be larger if the second largest eigenvalue of the adjacency matrix is larger.*
> > > >
> > > > *Additionally, Corollary 3.3 of [2] points out that the second largest eigenvalue of the adjacency matrix will be larger if a variable, which is named as commute time, is larger. Meanwhile, following Theorem 4.1 of [2], if we consider the input graph as an electrical network, the commute time is linear to the resistance between any two nodes in the graph.*
> > > >
> > > > *Based on the above pior arts' findings, our data-dependent dynamic skipping techniques can be regarded as removing the connections in the graph as illustrated in Eqs. 3-5 of our manuscript. Thus, the expectation of the resistance between any two nodes in the graph will be enlarged because removing the connections will be equivalent to removing the corresponding resistors: If those removed resistors formulate a parallel circuit together with other resistors, it will result in a larger resistance; If those removed resistors formulate a series circuit together with other resistors, it will result in an infinite resistance, which also corresponds to a larger resistance.*
> > > >
> > > > *So the above theoretical analysis illustrates that our data-dependent dynamic skipping techniques in the proposed D$^2$-GCN can be regarded as a special implicit regularization to enlarge the commute time [2] of the given graph, which can alleviate the over-smoothing issue via leading to a larger second largest eigenvalue of the adjacency matrix [2, 3].*
> > > >
> > > > We will add the above analysis and discussion to our final version, as it can help enhance readers' understanding and provide more insights about our proposed technique.
> > > >
> > > > We are happy to address any further comments or concerns, and wish you Happy Thanksgiving Holiday!
> > > >
> > > > *[1] Rong, Yu, et al. "Dropedge: Towards deep graph convolutional networks on node classification." arXiv preprint arXiv:1907.10903 (2019).*
> > > >
> > > > *[2] Lovász, László. "Random walks on graphs." Combinatorics, Paul erdos is eighty 2.1-46 (1993): 4*
> > > >
> > > > *[3] Oono, Kenta, and Taiji Suzuki. "Graph neural networks exponentially lose expressive power for node classification." arXiv preprint arXiv:1905.10947 (2019).*
> > > >
> > > > *[4] Chen, Ming, et al. "Simple and deep graph convolutional networks." International Conference on Machine Learning. PMLR, 2020.*

---

### Official Review · Reviewer_gZU6 · 2021-11-02

**Correctness:** 3
**Technical Novelty And Significance:** 4
**Empirical Novelty And Significance:** 4
**Recommendation:** 6
**Confidence:** 4

**Main Review:**


Strengths:
+ The paper is clearly written and easy to follow.
+  Integrating node-wise skepping, edge-wise skipping, an bit-wise skipping  to reduce the nodes, prune the edges, and compress the bit-precision is interesting and somewhat novel.
+ Experiments are extensive and promissing.


Weaknesses:

- In the discussion in Section 4.3 it is mentioned that "our proposed D$^2$GCN can resolve the over-smoothing issue." The reviewer is quite interesting how the over-smoothing issue is resolved in the proposed D$^2$GCN. More discussions, empirical or theoretical analysis on this point is needed.
- Furthermore, it is stated that  "such data-dependent skipping techniques serve as a regularization effect". It is appealing to mention of the regularization. However, the interpretation---" since different components of a graph may work together or independently in an data-dependent manner to avoid the over-smoothing issue" is disappointingly too vague.
Could it possible to develop any theoretical analysis on the skipping techniques to verify whether it is equivalent to a special implicit regularization and, what kind of impilicit regularization?   Since that the idea of the paper is very simple, it would make the paper stronger if the theory behind could be developed.


Minor issues:
There is a mistake in the subfigure of the right panel: $X_2 --> X_1$;


**Summary Of The Paper:**

The paper  attempts to address the training efficiency and scalability of GCNs. To be specific, a so-called Data-Dependent dynamic GCN framework is proposed, in which node-wise skepping, edge-wise skipping, an bit-wise skipping are integrated via gate function to squeeze out (or reduce) the unimportant neighbor nodes in combinations, unimportant edge connections, and in the bit-precision, respectively. Extensive experiments are provided, showing new SOTA results on benchmark datasets.

**Summary Of The Review:**

While the paper is clearly written and the empirical evaluation is convincing, it is not clear why and how the over-smoothing issue could be resolved or what kind of equivalent regularization behind.

---

> ### Author Response · Authors · 2021-11-23
> **Response to Reviewer gZU6**
>
> > + Q1: In the discussion in Section 4.3 it is mentioned that "our proposed D2-GCN can resolve the over-smoothing issue." The reviewer is quite interested in how the over-smoothing issue is resolved in the proposed D-2GCN. More discussions, empirical or theoretical analysis on this point is needed.
>
> A1: Thank you for your constructive suggestion! To better illustrate our proposed D$^2$-GCN can be helpful to alleviate the over-smoothing issue, we follow DropEdge [2] to use the distance of different intermediate layers’ output as the metric for measuring over-smoothing, under which the larger distance means the over-smoothing issue is less serious. We conduct experiments on both (1) the 64-layer GCNII model with the Cora dataset and (1) the 32-layer GCNII model with the Citeseer dataset, and summarize the results in the table below. The baseline is the vanilla GCNII model on the corresponding same datasets. From this set of experiments, we can observe that our proposed D$^2$-GCN outperforms GCNII in terms of alleviating the over-smoothing issue, e.g., enlarging the distance of different intermediate layers by 1.38x ~ 2.40x, while this improvement is difficult to be observed from merely the accuracy (e.g., +0.0% ~ +0.1% as compared to GCNII).
>
> We will include this set of experiments and corresponding discussions in our final version.
>
> |Method | Model@Dataset | Accuracy (%) | Avg. Distance After Training|
> |:-:|:-:|:-:|:-:|
> |Baseline | GCNII@Cora | 85.5 | 1.59 |
> | **Ours** | GCNII@Cora | **85.6 (+0.1)** | **2.19 (1.38x)** |
> | Baseline | GCNII@Citeseer | 73.4 | 1.27 |
> | **Ours** | GCNII@Citeseer | **73.4 (+0.0)** | **3.05 (2.40x)** |
>
> > + Q2: Furthermore, it is stated that "such data-dependent skipping techniques serve as a regularization effect". It is appealing to mention the regularization. However, the interpretation---" since different components of a graph may work together or independently in an data-dependent manner to avoid the over-smoothing issue" is disappointingly too vague. Could it be possible to develop any theoretical analysis on the skipping techniques to verify whether it is equivalent to a special implicit regularization and, what kind of implicit regularization? Since the idea of the paper is very simple, it would make the paper stronger if the theory behind it could be developed.
>
> A2: Thank you for the suggestion! We provide a brief theoretical analysis below following [the theory of random walks on graphs](https://web.cs.elte.hu/~lovasz/erdos.pdf) [2].
>
> As mentioned by DropEdge [1], the over-smoothing issue can be measured by the distance of different intermediate layers’ output [1]. A larger distance means that the over-smoothing issue is less serious. As proved in Theorem 1 of [3], the upper bound of such a distance will be larger if the second largest eigenvalue of the adjacency matrix is larger.
>
> Additionally, Corollary 3.3 of [2] points out that the second largest eigenvalue of the adjacency matrix will be larger if a variable, which is named as commute time, is larger. Meanwhile, following Theorem 4.1 of [2], if we consider the input graph as an electrical network, the commute time is linear to the resistance between any two nodes in the graph.
>
> Based on the above pior arts' findings, our data-dependent dynamic skipping techniques can be regarded as removing the connections in the graph as illustrated in Eqs. 3-5 of our manuscript. Thus, the expectation of the resistance between any two nodes in the graph will be enlarged because removing the connections will be equivalent to removing the corresponding resistors: If those removed resistors formulate a parallel circuit together with other resistors, it will result in a larger resistance; If those removed resistors formulate a series circuit together with other resistors, it will result in an infinite resistance, which also means larger resistance.
>
> So the above theoretical analysis illustrates our data-dependent dynamic skipping techniques can be regarded as a special implicit regularization to enlarge the commute time [2] of the given graph, which can alleviate the over-smoothing issue via a larger second largest eigenvalue of the adjacency matrix [2, 3].
>
> We will add the above analysis and discussion to our final version, as it can help enhance readers' understanding and provide more insights about our proposed technique.
>
> > Q3: Minor issues
>
> A3: Thank you for pointing it out! We have corrected them in the updated manuscript.
>
>
> *[1] Rong, Yu, et al. "Dropedge: Towards deep graph convolutional networks on node classification." arXiv preprint arXiv:1907.10903 (2019).*
>
> *[2] Lovász, László. "Random walks on graphs." Combinatorics, Paul erdos is eighty 2.1-46 (1993): 4.*
>
> *[3] Oono, Kenta, and Taiji Suzuki. "Graph neural networks exponentially lose expressive power for node classification." arXiv preprint arXiv:1905.10947 (2019).*

---

> > ### Comment · Reviewer_gZU6 · 2021-11-29
> > **Reply to the responsesin the rebuttal**
> >
> > While the responses in the rebuttal have partially addressed the concerns in the review, the reviewer is still not satisfactory to the answers the over-smoothing issue and the implicit regularization interpretations.

---

### Official Review · Reviewer_u2pj · 2021-11-02

**Correctness:** 3
**Technical Novelty And Significance:** 2
**Empirical Novelty And Significance:** 2
**Recommendation:** 6
**Confidence:** 4

**Main Review:**

I think D2-GCN is well-designed with simple components and the experimental results are impressive in that many recent graph data are at large-scale (especially, web data).

The authors focus on the efficiency of computation but I want to ask about the effectiveness of information processing.

1. Authors argue that it can be helpful to alleviate the over-smoothing issue. It makes sense because three gating (or blocking) functions might prevent mixing node information. However, is there any experimental evidence (e.g. measurement of smoothness)?

2. Can it be extended as (integrated with) a disentangled representation learning framework for GNNs? If so, could you briefly explain it? Because cutting connection and quantizing information seem like Disentangled GCN [1] in broad concept. I think it would be interesting to analyze which information is discarded or not.


The above questions are just for constructive discussion.


[1] Disentangled Graph Convolutional Networks (ICML 2019)

**Summary Of The Paper:**

In this work, the authors propose relatively low-cost GCNs in a data-dependent way.

Their framework has three main components: node-wise, edge-wise, and bit-wise skipping.

1. Node-wise skipping is determined by a binary decision for each node based on its features.

2. Edge-wise skipping is about the removal of connections between two nodes.

3. Bit-wise skipping is about the quantization precision of aggregated node features.


It boosts efficiency while achieving comparable performance over benchmark datasets.

**Summary Of The Review:**

I would recommend this paper to be accepted if other issues do not arise.

---

> ### Author Response · Authors · 2021-11-23
> **Response to Reviewer u2pj**
>
> > + Q1: Authors argue that it can be helpful to alleviate the over-smoothing issue. It makes sense because three gating (or blocking) functions might prevent mixing node information. However, is there any experimental evidence (e.g. measurement of smoothness)?
>
> A1: Thank you for your constructive suggestion! To better illustrate our proposed D$^2$-GCN can be helpful to alleviate the over-smoothing issue, we follow DropEdge [2] to use the distance of different intermediate layers’ output as the metric for measuring the over-smoothing level, under which the larger distance means the over-smoothing issue is less serious. We conduct the experiments on both the 64-layer GCNII model with the Cora dataset and the 32-layer GCNII model with the Citeseer dataset and summarize the results in the table below. The baseline is the vanilla GCNII model on the corresponding same datasets. We can observe that our proposed D$^2$-GCN outperforms GCNII in terms of alleviating the over-smoothing issue, e.g., enlarging the distance of different intermediate layers by 1.38x ~ 2.40x, while this improvement is difficult to be observed from merely the accuracy (e.g., +0.0% ~ +0.1% as compared to GCNII).
>
> We will include this set of experiments and corresponding discussions in our final version.
>
> |Method | Model@Dataset | Accuracy (%) | Avg. Distance After Training|
> |:-:|:-:|:-:|:-:|
> |Baseline | GCNII@Cora | 85.5 | 1.59 |
> | **Ours** | GCNII@Cora | **85.6 (+0.1)** | **2.19 (1.38x)** |
> | Baseline | GCNII@Citeseer | 73.4 | 1.27 |
> | **Ours** | GCNII@Citeseer | **73.4 (+0.0)** | **3.05 (2.40x)** |
>
> > + Q2: Can it be extended as (integrated with) a disentangled representation learning framework for GNNs? If so, could you briefly explain it? Because cutting connections and quantizing information seem like Disentangled GCN [1] in a broad concept. I think it would be interesting to analyze which information is discarded or not.
>
> A2: Thank you for proposing this interesting perspective to view our proposed D$^2$-GCN! Yes, our proposed D$^2$-GCN can be extended as (integrated with) a disentangled representation learning framework for GNNs.
>
> The DisenGCN introduced in [1] dynamically identifies the latent factor that may have caused the edge between a node and one of its neighbors to form a neighborhood routing mechanism, which is proven to be very effective to boost the accuracy of GCN in multiple graph tasks. Meanwhile, our proposed D$^2$-GCN performs data-dependent dynamic skipping at multiple granularities with the goal of both better accuracy vs. inference efficiency trade-off and better trainability of deeper GCNs. So it is possible to extend our proposed D$^2$-GCN as a disentangled representation learning framework for GNNs by adding extra regularizations to make the graph after skipping form different clusters so that each node is more disentangled to each other, and thus further boost the accuracy of GCNs by leveraging the effectiveness of the disentangled representation learning.
>
> *[1] Disentangled Graph Convolutional Networks (ICML 2019)*
>
> *[2] Rong, Yu, et al. "Dropedge: Towards deep graph convolutional networks on node classification." arXiv preprint arXiv:1907.10903 (2019).*

---

### Official Review · Reviewer_pLUm · 2021-11-03

**Correctness:** 3
**Technical Novelty And Significance:** 3
**Empirical Novelty And Significance:** 3
**Recommendation:** 5
**Confidence:** 3

**Main Review:**


Strength:

The writing is easy to understand and the idea is clearly presented. The experiment results show good empirical inference speed compared to the existing methods.


Weakness&Advice:

1. The description of “The inference process of one GCN layer can be viewed as two separated phases: Aggregation and Combination. …” for equation (1) is not precise. See page 52 in [1]. Instead of being viewed as two separated phases, “we no longer need to define an explicit update function, as the update is implicitly defined through the aggregation method”.

2. “we hypothesize that explicit or implicit regularizations during GCN training can improve GCNs’ scalability towards going deeper,”  Randomly dropping being able to help with deeper GCN does not mean all regularization methods can. This hypothesis is over-claimed and groundless.

3. What is the training speed of your model and the three-stage training pipeline compared to the existing methods?

4. "$L_{comp}$  is the computational cost determined by the decisions made by the gating functions” What does this sentence means? It’s better to write $\mathcal{L}_{comp}$ in formula.

5. What is the performance after each stage in your three-stage training pipeline. I would like to know which stage plays the most important role and how other stages improve the performance.

6. Gating is not a new technique for GNNs, e.g. [2]. It’s better to draw the difference with these methods and emphasize the novelty and contribution of your method. Note that I will not consider it novel if you only put some existing tricks into your model.

Typos:
Section 3.2, 3rd paragraph, graphes —> graphs

Bit-wise skipping in figure 1, $x_2$ —> $x_1$

In equation (4), should use parenthesis after $\odot$ to avoid confusion.

[1] Hamilton WL. Graph representation learning. Synthesis Lectures on Artifical Intelligence and Machine Learning. 2020 Sep 15;14(3):1-59.

[2] Li Y, Tarlow D, Brockschmidt M, Zemel R. Gated graph sequence neural networks. arXiv preprint arXiv:1511.05493. 2015 Nov 17.



**Summary Of The Paper:**

The authors propose a Data-Dependent GCN framework D$^2$-GCN which integrates data-dependent node-wise, edge-wise and bit-wise skipping to save the cost for inference while does not sacrifice prediction accuracy.

**Summary Of The Review:**

Some of the technical details need to be provided. The novelty need to be emphasized. I'll consider raise my score if my concerned can be properly addressed.

---

> ### Author Response · Authors · 2021-11-23
> **Response to Reviewer pLUM**
>
> > + Q1: The description of “The inference process of one GCN layer can be viewed as two separated phases: Aggregation and Combination. …” for equation (1) is not precise. See page 52 in [1]. Instead of being viewed as two separated phases, “we no longer need to define an explicit update function, as the update is implicitly defined through the aggregation method”.
>
> A1: Thank you for your suggestions! We have modified the corresponding description in the updated manuscript to point out the case mentioned in [1].
>
> > + Q2: “We hypothesize that explicit or implicit regularizations during GCN training can improve GCNs’ scalability towards going deeper,” Randomly dropping being able to help with deeper GCN does not mean all regularization methods can. This hypothesis is over-claimed and groundless.
>
> A2: Thank you for the suggestions! We have modified it in the updated manuscript.
>
> > + Q3: What is the training speed of your model and the three-stage training pipeline compared to the existing methods?
>
> A3: We measure the training time in an NVIDIA GeForce RTX 2080 Ti GPU with Pytorch Geometric framework and summarize the training time in the table below. The baseline is the 7-layer DeeperGCN model on ogbg_molhiv dataset and we add our proposed D$^2$-GCN on top of it. We can have the following observations: 1) our proposed D$^2$-GCN costs \~3x of the baseline training time but saves \~17X inference FLOPs with nearly the same accuracy; 2) the stage 1 of D$^2$-GCN is the same with the baseline, so it can be omitted if we can leverage the public available checkpoints and reduce the training cost from \~3x to only \~2x of that in the baseline training; 3) the stage 2 of D$^2$-GCN only takes a small portion of the total time (\~0.3%) because only the gating function weights need to be trained in this stage; 4) the stage 3 of D$^2$-GCN is the most time consuming stage as it needs to train both the weights of GCN and gating functions with the goal of balancing the accuracy and inference FLOPs.
>
> |Training Method | Training Time | Accuracy (%) | FLOPs (G)|
> |:-:|:-:|:-:|:-:|
> |Baseline | 6.50 hours | 78.6 | 483.14|
> |Ours | 19.05 hours | 78.5 | 28.73|
> |Ours - Stage 1 | 6.50 hours| - | - |
> |Ours - Stage 2 | 0.05 hours | - | - |
> |Ours - Stage 3 | 12.50 hours | - | - |
>
> > + Q4: $\mathcal{L}_{comp}$ is the computational cost determined by the decisions made by the gating functions” What does this sentence mean? It’s better to write ‘Lcomp’ in formula.
>
> A4: Thank you for your suggestion! We have included a detailed formula in Eq. 7 of the updated manuscript to show how the $\mathcal{L}_{comp}$  is determined by the gating functions.
>
> > + Q5: What is the performance after each stage in your three-stage training pipeline? I would like to know which stage plays the most important role and how other stages improve the performance.
>
> A5: Thank you for your suggestions! To illustrate the performance after each stage, we conduct experiments in the 7-layer DeeperGCN models on ogbg_molhiv datasets and summarize the results in the table below. We can observe that stage 3 plays the most important role because it jointly trains the weights of GCN and gating functions to balance the accuracy and inference FLOPs. But other two stages are also important because they can be regarded as generating a decent initialization for the GCN weights and the gating function weights, respectively, as illustrated in Section 3.4. Without the first two stages, the joint training process in stage 3 is not convergent.
>
> | Stage Index | Accuracy (%) | FLOPs (G)|
> |:-:|:-:|:-:|
> |1 | 0.786 | 483.14 |
> |2 | 0.454 | 25.91 |
> |3 | 0.785 | 28.73 |

---

> > ### Author Response · Authors · 2021-11-23
> > **Response to Reviewer pLUM**
> >
> > > + Q6: Gating is not a new technique for GNNs, e.g. [2]. It’s better to draw the difference with these methods and emphasize the novelty and contribution of your method. Note that I will not consider it novel if you only put some existing tricks into your model.
> >
> > A6: Thanks for your suggestions! We humbly justify that our novelty/contributions as compared with previous gating or dropping techniques of GCNs (e.g., [2, 3]) below are nontrivial: 1) we focus on **boosting both the accuracy vs. efficiency trade-off and the trainability** of deeper GCNs, while previous gating or dropping techniques only focus on the latter, e.g., [2] uses RNN and LSTM as the gating function without considering the overhead of their heavy gating functions may be larger than the GCNs themselves and [3] is only applicable to the training period without reducing the inference cost; 2) our work for the first time proposes, designs, and validates the feasibility to **squeeze the redundancy of GCNs simutaneously from multiple granularities**: node-wise, edge-wise, and bit-wise, while previous works only focus on node-wise [2] or edge-wise [3] only, and all these granularities are **dedicatedly designed for GCNs** but not just putting some existing tricks from CNNs [4,5] into GCNs because those granularities are coupled with the input of graph format instead of image format; and 3) technically our work **contributes a new systematic way to remove the redundancy of GCNs from all/various dimensions with a low overhead**, leading to large savings in GCN inference.
> >
> > > + Q7: Typos
> >
> > A7: Thank you for pointing them out! We have corrected them in the updated manuscript.
> >
> >
> > *[1] Hamilton WL. Graph representation learning. Synthesis Lectures on Artificial Intelligence and Machine Learning. 2020 Sep 15;14(3):1-59.*
> >
> > *[2] Li Y, Tarlow D, Brockschmidt M, Zemel R. Gated graph sequence neural networks. arXiv preprint arXiv:1511.05493. 2015 Nov 17.’*
> >
> > *[3] Rong, Yu, et al. "Dropedge: Towards deep graph convolutional networks on node classification." arXiv preprint arXiv:1907.10903 (2019).*
> >
> > *[4] Wang, Xin, et al. "Skipnet: Learning dynamic routing in convolutional networks." Proceedings of the European Conference on Computer Vision (ECCV). 2018.*
> >
> > *[5] Wang, Yue, et al. "Dual dynamic inference: Enabling more efficient, adaptive, and controllable deep inference." IEEE Journal of Selected Topics in Signal Processing 14.4 (2020): 623-633.*

---

### Decision · Program_Chairs · 2022-01-20

**Decision:**

Reject

**Comment:**

The paper proposes Data-Dependent GCN (D2-GCN), which improves the efficiency of vanilla GCN by node-wise skipping, edgewise skipping, and bit-wise skipping. Gate functions are learned to prune the unimportant neighbor nodes in combinations, unimportant edge connections, and in the bit-precision. The proposed method boosts efficiency while achieving comparable performance over benchmark datasets. Most reviewers agree that the paper is well motivated, and the writing is clear. However, two of the reviewers found the novelty of the paper compared to previous work (for example, [1]) is limited. Three reviewers raised concerns about the lack of theoretical or empirical analysis on how D2-GCN can alleviate the over-smoothing problem, and how the proposed method can serve as an implicit regularization.

For the novelty concerns, the authors provided a detailed comparison with previous work during the rebuttal. For the lack of analysis on over-smoothing, the authors provided an additional empirical analysis using the distance of different intermediate layers’ output as the metric for measuring over-smoothing. But at least one reviewer is still not satisfied with those.

Given the current review scores (3, 5, 5, 6), the paper is below the acceptance threshold for the conference. The AC believes that the proposed method seems to be an effective and simple way towards more efficient graph neural networks and hence encourages the authors to submit the revised paper to another venue after addressing the reviewers’ concerns, especially on theoretical or empirical analysis on over-smoothing and implicit regularization.


[1]: Gated graph sequence neural networks